# Budget-Aware Sequential Brick Assembly with Efficient Constraint Satisfaction

**Seokjun Ahn**[*]  *sdeveloper@postech.ac.kr*
*POSTECH*

**Jungtaek Kim**[*]  *jungtaek.kim@pitt.edu*
*University of Pittsburgh*

**Minsu Cho**  *mscho@postech.ac.kr*
*POSTECH*

**Jaesik Park**  *jaesik.park@snu.ac.kr*
*Seoul National University*

**Reviewed on OpenReview:** *https://openreview.net/forum?id=0mRfQOnkqk*

## Abstract

We tackle the problem of *sequential brick assembly* with LEGO bricks to create combinatorial 3D structures. This problem is challenging since this brick assembly task encompasses the characteristics of combinatorial optimization problems. In particular, the number of assemblable structures increases exponentially as the number of bricks used increases. To solve this problem, we propose a new method to predict the scores of the next brick position by employing a U-shaped sparse 3D convolutional neural network. Along with the 3D convolutional network, a *one-initialized brick-sized* convolution filter is used to efficiently validate assembly constraints between bricks without training itself. By the nature of this one-initialized convolution filter, we can readily consider several different brick types by benefiting from modern implementation of convolution operations. To generate a novel structure, we devise a sampling strategy to determine the next brick position considering the satisfaction of assembly constraints. Moreover, our method is designed for either *budget-free* or *budget-aware* scenario where a budget may confine the number of bricks and their types. We demonstrate that our method successfully generates a variety of brick structures and outperforms existing methods with Bayesian optimization, deep graph generative model, and reinforcement learning.

## 1 Introduction

Most real-world 3D structures are constructed with smaller *primitives*. A broad range of studies have tackled interesting assembly problems such as molecule generation (Ertl et al., 2017; Neil et al., 2018; You et al., 2018), building construction (Talton et al., 2011; Martinovic & Van Gool, 2013; Ritchie et al., 2015), and assembly generation (Sung et al., 2017; Lee et al., 2021; Jones et al., 2021b; Lee et al., 2022).[1] In particular, if unit primitives are used to construct 3D structures we desire to assemble, this task becomes an instance of combinatorial optimization problems, in which a search space increases exponentially as a search depth increases. Formally, given $n$ primitives with $k$ possible connections between two primitives, the search space increases by $\mathcal{O}(k^n)$. In addition to the combinatorial property, the consideration of assembly constraints between unit primitives makes the problem more challenging.

---

[*]Equal contribution.
[1]The implementation of our method is available at `https://github.com/joonahn/BrECS`.

Table 1: Comparisons of the existing approaches and our method. SA, BayesOpt, GGM, RL, and SL stand for sequential assembly, Bayesian optimization, graph generative model, reinforcement learning, and supervised learning, respectively.

| Method | Unit Primitives | Algorithm | Target Conditioning | Constraint Satisfaction |
|---|---|---|---|---|
| SA w/ BayesOpt | Single | BayesOpt | Target volume | Subsampling |
| DGMLG | Single | GGM | Class label | Masking |
| Brick-by-Brick | Single | RL | Single- or multi-view images | SL |
| BrECS (Ours) | Multiple | SL | No target conditioning for a generation task
Incomplete target volume for a completion task | Convolution operations |

The problem of sequential assembly with LEGO bricks inherits the aforementioned properties, in that a decision-making process sequentially determines where a brick is placed. Supposing that we are given many bricks to assemble, there are a large number of assemblable combinations. Moreover, the decision-making process must consider complex assembly constraints, i.e., the disallowance of overlap, no isolated bricks, and LEGO-specific connections. Compared to generic 3D generation methods (Wu et al., 2016; Gadelha et al., 2018; Achlioptas et al., 2018), our brick assembly task can generate 3D structures in an open space and provide brick-wise instructions to build the structures.

Several attempts have been employed to solve sequential brick assembly by utilizing Bayesian optimization (Kim et al., 2020), deep graph generative models (Thompson et al., 2020), and reinforcement learning (Chung et al., 2021), as summarized in Table 1. Those methods end with the consideration of the assembly with only $2 \times 4$ LEGO bricks – they might be capable of assembling other brick types without significant modification, though. Moreover, the previous literature has several respective limitations. The Bayesian optimization-based method (Kim et al., 2020) requires heavy computations due to its iterative optimization process. Another method (Thompson et al., 2020) has been proposed using masks to filter out invalid actions along with their graph generative model, but the use of masks degrades assembly performance. To predict a valid action, the recent work (Chung et al., 2021) utilizes an auxiliary neural network that often fails to predict legitimate moves perfectly. More importantly, these methods share a common limitation: *they inevitably become slower in validating assembly constraints as the number of bricks increases*, due to the exponentially increasing search spaces.

To tackle the limitations mentioned above, we devise a novel brick assembly method with a U-shaped neural network utilizing a *one-initialized brick-sized* convolution filter to validate complex constraints efficiently. Notably, our one-initialized convolution filter enables our method to validate the constraints in a parallelizable and scalable manner without training itself, where several different brick types are considered. Using the sampling procedure proposed in this work, our method can create diverse sequences of LEGO bricks to generate high-fidelity brick structures. Furthermore, we consider two scenarios, *budget-free* and *budget-aware* assembly pipelines where a budget limits the number of bricks and their types. Henceforth, we refer to our method as sequential **Br**ick assembly with **E**fficient **C**onstraint **S**atisfaction (BrECS).

In summary, the contributions of this work are as follows:

- We propose a novel sequential brick assembly model, which validates assembly constraints with a one-initialized brick-sized convolution filter and generates high-fidelity 3D structures;

- We utilize a U-shaped sparse 3D convolutional neural network that is trained with a voxelized dataset of ModelNet40 (Wu et al., 2015);

- We show that our model successfully assembles different brick types in various circumstances, including a budget-aware scenario.

## 2 Related Work

**3D Shape Generation.** Point cloud and voxel representations are widely used in 3D shape generation. Various methods with variational auto-encoders (Gadelha et al., 2018), generative adversarial networks (Achlioptas et al., 2018), and normalizing flows (Yang et al., 2019) have been proposed to generate point clouds. On the other hand, voxel-based 3D shape generation has been studied. Wu et al. (2016) propose a generative adversarial network to generate voxel occupancy. Choy et al. (2019) generate 3D shapes on voxel grids with a fully convolutional network that includes efficient sparse convolution. Zhang et al. (2021) propose a method to generate high-quality voxel-based shapes applying 3D convolutional networks.

**Sequential Part Assembly.** Similar to the sequential brick assembly problem, sequential part assembly shares common difficulties with a combinatorial optimization problem. Chen et al. (2022) introduce a pairwise shape assembly model but the model is limited to the pairwise assembly. Jones et al. (2021a); Willis et al. (2022) suggest CAD parts assembly model which assembles parts by exploiting semantic CAD shape information and extracting features from it. Ghasemipour et al. (2022) aim to assemble multi-part objects and tackle the problem with large-scale reinforcement learning and graph-based model architecture. Sung et al. (2017) assemble incomplete 3D parts with a part retrieval network and a position prediction network. Hu et al. (2020) propose a model that sequentially moves a piled box into another container with different shapes. They tackle the problem by representing previous boxes into graphs and reinforcement learning with rewards considering physical constraints in a new container.

**Sequential Brick Assembly.** Unlike generic 3D shape generation methods and part assembly methods, the sequential brick assembly approaches (Kim et al., 2020; Thompson et al., 2020; Chung et al., 2021) consider the assembly constraints that are introduced by attachable connections between two adjacent bricks and the disallowance of brick overlap, as shown in Figure 2. As discussed in the work (Kim et al., 2020), these constraints encourage us to accentuate the nature of combinatorial optimization since a huge number of assemblable combinations exist in the presence of complex constraints. However, it is not trivial to validate such constraints. As in Table 1, to overcome these difficulties, Kim et al. (2020) sample a subset of available next brick positions, Thompson et al. (2020) mask out invalid positions by validating all possible positions, and Chung et al. (2021) train an auxiliary network for validating positions.

**Brick Assembly in Physical World.** Luo et al. (2015) show a method to find LEGO brick structures in consideration of physical constraints. Nägele et al. (2020) propose a two-layer planning approach for multi-robot LEGO brick assembly. Li et al. (2021) suggest a bi-level robot framework to learn to design and build bridges with blocks under the assumption that a blueprint is not accessible. Liu et al. (2023) develop a robot system to learn assembly and disassembly processes from human demonstration. Liu et al. (2024) provide analysis on the physical stability of 3D structures with LEGO bricks examining force balancing equations.

## 3 Budget-Aware Sequential Brick Assembly

In this section we introduce the brick assembly problem tackled in this paper.

**Formulation.** Similar to the previous work (Chung et al., 2021), a new brick is connected to one or (possibly) more bricks of previously assembled bricks. It implies that the position of the next brick can be determined as a relative displacement from a certain brick directly connected to the next brick. we denote this indicator brick as a *pivot brick*. After choosing the pivot brick, we finally determine a relative position to assemble from the pivot brick. Ultimately, the goal of this work is to predict a sequence of brick placements from scratch or from an incomplete brick structure. In particular, in an inference stage, our model does not utilize any guidance for final brick structures.

**Objectives.** Our sequential brick assembly aims at assembling LEGO bricks to 3D structures that resemble the 3D structures used in the training of our proposed model. As described above, we do not provide any guidance for final target structures in an inference stage and construct 3D structures from scratch or from incomplete brick structures. Note that a scenario of building 3D structures from scratch is called a *generation*

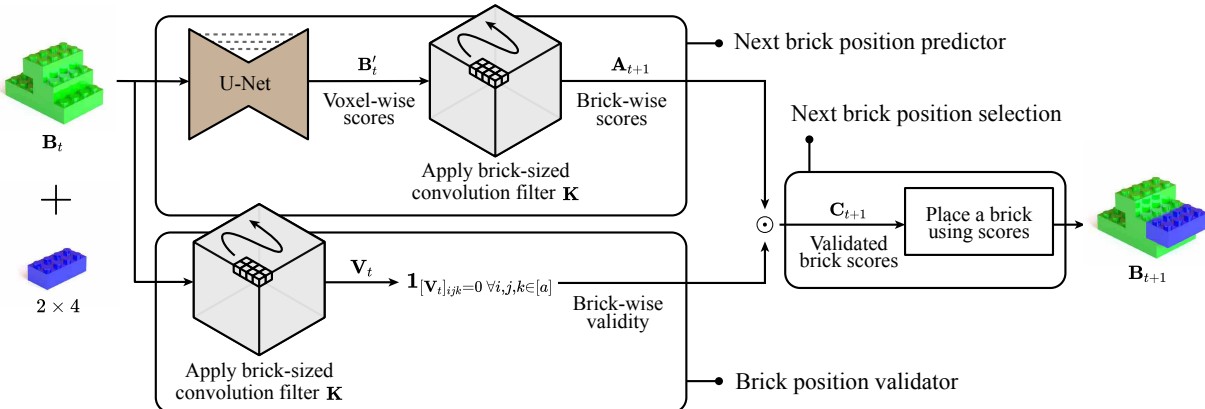

Figure 1: Illustration of the efficient constraint satisfaction method with convolution filters for sequential brick assembly. U-Net and $\odot$ denote a U-shaped neural network with sparse 3D convolutional layers and element-wise multiplication, respectively.

*task* and a scenario of building 3D structures from incomplete brick structures is called a *completion task*. Moreover, we consider a budget-free or budget-aware scenario in sequential brick assembly.

**Constraints.** Inspired by the standard LEGO bricks, we take into account three constraints in the brick assembly problem: (i) bricks should not overlap with each other (NO-OVERLAP); (ii) all bricks of the current structure should be connected to each other so that the current structure is represented as one connected structure (NO-ISOLATION); (iii) a new brick must be directly attached to the upper or lower position of other bricks (VERTICAL-ASSEMBLE).

**Budgets.** Suppose that a brick budget is predefined, where a budget indicates the number of assemblable bricks for each brick type. For example, if we are given four $2 \times 4$ and two $2 \times 2$ LEGO bricks as a brick budget, we can only assemble those six bricks in total. On the other hand, if we are given an infinite budget, our framework can choose any brick type without restriction.

## 4 Proposed Approach

Here, we explain four steps of our method BrECS, which are illustrated in Figure 1. To sum up, we generate a brick structure under assembly constraints by repeating the following steps: (i) score computation of next brick positions, (ii) exclusion of invalid positions, (iii) sampling of a pivot brick, and (iv) determination of a relative brick position. Note that a neural network for computing the scores of next brick positions is an *only learnable* component in our framework.

### 4.1 Efficient Constraint Satisfaction

We propose a novel method to tackle the challenge of satisfying the following constraints: NO-OVERLAP, which is validated by using convolution operations; NO-ISOLATION and VERTICAL-ASSEMBLE, which are satisfied by following the brick assembly formulation with pivot bricks and relative brick positions. Borrowing the concept of constraint satisfaction (Tsang, 1993), which is the problem of finding solutions that satisfy a predefined set of constraints, our method is designed to establish an approach to efficient constraint satisfaction for sequential brick assembly.

**Predicting Next Brick Positions.** Given a voxel representation of a structure at step $t$, which is denoted as $\mathbf{B}_t \in \{0, 1\}^{a \times a \times a}$ where $a$ is the size of 3D space, we first feed the voxel representation $\mathbf{B}_t$ into a U-shaped

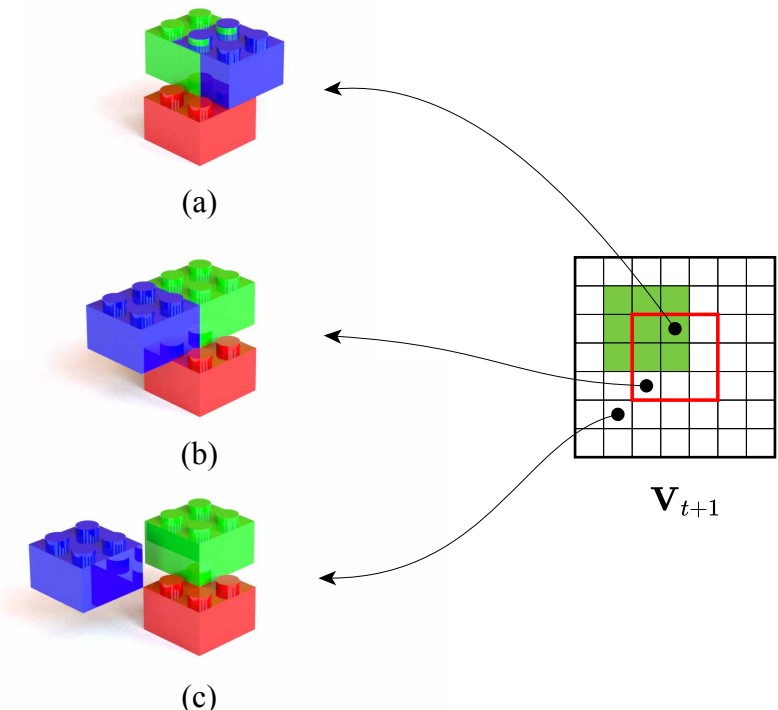

Figure 2: Illustration of how one-initialized brick-size convolution filters work where a red brick is a pivot brick and a green brick has been assembled to the red brick. The grid shows the validity $\mathbf{V}_{t+1}$ of each brick position on the red brick for a brick type $2\times2$. The pixels in a red rectangle indicate attachable brick positions on the red brick. The green pixels indicate that overlap with the green brick will occur if a new brick is attached to the corresponding position. Therefore, (a) and (c) violate NO-OVERLAP and NO-ISOLATION, respectively, and (b) satisfies the assembly constraints.

sparse 3D convolutional neural network, inspired by Choy et al. (2019):

$$\mathbf{B}'_t = \text{U-Net}(\mathbf{B}_t), \tag{1}$$

in order to capture global and local contexts effectively and retain the same dimensionality. Due to its pyramidal feature extraction structure, the U-Net extracts robust features understanding multi-dimensional contexts. In particular, we validate that the U-Net effectively extracts important contexts and thus improves overall performance compared to other neural networks; see Table 6 for a thorough study on neural architectures. Moreover, we expect that our neural network produces a likely complete or potentially next-step 3D structure, which is represented by a probability of voxel occupancy. Note that the network parameters in this U-Net are the only learnable component in our framework BrECS. In this section, we assume that we are given the pretrained U-Net.

After obtaining $\mathbf{B}'_t \in \mathbb{R}^{a\times a\times a}$, scores for next brick positions $\mathbf{A}_{t+1} \in \mathbb{R}^{a\times a\times a}$ are computed by sliding a convolution filter $\mathbf{K} \in \mathbb{R}^{w_b\times d_b\times 1}$ across $\mathbf{B}'_t$:

$$\mathbf{A}_{t+1} = \mathbf{B}'_t * \mathbf{K}, \tag{2}$$

where $*$ is a convolution operation. We match the size of $\mathbf{A}_{t+1}$ to the size of $\mathbf{B}_t$ by applying zero padding. In particular, the size of the convolution filter is the same as the brick size we assemble $w_b \times d_b$ where $w_b$ and $d_b$ are the width and depth of the brick, respectively, so that we can determine the scores over all the possible positions of the next brick by aggregating the corresponding voxels. For example, if we use $2\times4$ bricks, the size of the convolution filter is $2 \times 4 \times 1$. Moreover, $\mathbf{K}$ is always initialized as a tensor filled with 1 without updating its values in a training stage to aggregate $w_b d_b$ voxels equally with a single convolution operation.

In the case of assembling a structure with $r$ brick types, we use $2r$ convolution filters of the same size with $r$ brick types and repeat the above process for each convolution filter. Notably, for each brick type, we employ two convolutional filters (i.e., $w_b \times d_b \times 1$ and $d_b \times w_b \times 1$) to consider brick rotation.

**Filtering out Invalid Brick Positions.** As shown in Figure 1, to predict the validity of the next brick positions, a one-initialized brick-sized convolution filter is applied to $\mathbf{B}_t$:

$$\mathbf{V}_t = \mathbf{B}_t * \mathbf{K}. \tag{3}$$

The filter $\mathbf{K}$ is identical to the filter used in computing the score for the next brick positions. After applying the convolution filter across $\mathbf{B}_t$, all attachable brick positions are determined if the value of the position of interest is zero, which means that no overlap exists within the brick positions. As visualized in Figure 2, the validity of each position in terms of overlap and the number of attachable positions with respect to the brick of interest are readily determined by applying the convolution filter of the corresponding brick type.

Using these two branches for computing $\mathbf{A}_{t+1}$ and $\mathbf{V}_t$, which are shown in Equations (2) and (3), we can define validated scores for next brick positions $\mathbf{C}_{t+1}$. Each entry of $\mathbf{C}_{t+1}$ is equal to its score for next brick positions filtering out invalid brick positions. Formally, we describe $\mathbf{C}_{t+1}$ of assembling a $w_b \times d_b$ brick on a pivot brick:

$$\begin{aligned}
\mathbf{C}_{t+1} &= \mathbf{1}_{[\mathbf{V}_t]_{ijk}=0 \ \forall i,j,k \in [a]}(\mathbf{V}_t) \odot \mathbf{A}_{t+1} \\
&= \mathbf{1}_{[\mathbf{B}_t*\mathbf{K}]_{ijk}=0 \ \forall i,j,k \in [a]}(\mathbf{B}_t * \mathbf{K}) \odot (\text{U-Net}(\mathbf{B}_t) * \mathbf{K}),
\end{aligned} \tag{4}$$

where $\mathbf{1}_{[\mathbf{V}_t]_{ijk}=0 \ \forall i,j,k \in [a]}$ is an indicator function, $\mathbf{K} \in \mathbb{R}^{w_b \times d_b \times 1}$ is the one-initialized convolution filter, and $*$ is a convolution operation. Since the calculation of the masked score tensor $\mathbf{C}_{t+1}$ uses convolution operations, we efficiently compute it with modern GPU devices. Specifically, while the validity check of $(64, 64, 64)$ voxels without parallelization takes 4.7 seconds, the identical validity check with our method takes only 0.0021 seconds; ours yields approximately *2,300 times faster* inference on the validity check.

**Selecting Pivot Bricks by Sampling.** To place a single brick, we need to choose one of previously assembled bricks, i.e., a pivot brick, to attach a new brick. The motivation of our method to select a pivot brick is that a pivot brick with higher sum of attachable brick scores should be preferable to one with lower sum of attachable brick scores, rather than choosing a pivot brick that is connected to a position with the highest score of $\mathbf{C}_{t+1}$. Since a neural network tends to memorize training samples and their assembly sequences, choosing a position with the highest score fails to create a novel structure. Instead of a deterministic approach based on $\mathbf{C}_{t+1}$, we alter a method to select a pivot brick into a sampling method. To compare the number of attachable positions, we define a pivot score $T_{ijk}$ of the pivot of $(i, j, k)$ to aggregate scores of attachable positions:

$$T_{ijk} = \sum_{l=i-(\lfloor w' \rfloor - 1)}^{i+(\lceil w' \rceil - 1)} \sum_{m=j-(\lfloor d' \rfloor - 1)}^{j+(\lceil d' \rceil - 1)} \sum_{n \in \{k-1, k+1\}} \mathbf{C}_{lmn}, \tag{5}$$

where $w' = \frac{w_b + w_p}{2}$, $d' = \frac{d_b + d_p}{2}$, and $w_p \times d_p$ is the size of a pivot brick candidate. After computing pivot scores, we employ a sampling strategy to determine a pivot brick:

$$(a, b, c) \sim \mathbf{p}, \tag{6}$$

where

$$[\mathbf{p}]_{ijk} = \frac{T_{ijk}}{\sum_{(l,m,n) \in \text{pivots}} T_{lmn}}. \tag{7}$$

Our sequential procedure for pivot brick predictions is inspired by Monte-Carlo tree search (MCTS) (Coulom, 2006). MCTS evaluates possible actions by expanding a search tree with Monte-Carlo simulations and backups. Similar to this, our method also evaluates pivot brick candidates by aggregating their attachable brick scores, so that pivot with more attachable bricks will more likely be selected. Our method utilizes the U-shaped sparse 3D convolutional neural network to predict possible brick positions, which is analogous to a policy network of the MCTS method (Silver et al., 2016), which employs a neural network to predict prior distributions for search tree expansion.

---

**Algorithm 1** Assembly of a single brick

---

**Input:** Voxels of structure at current time step $\mathbf{B}$, a list of assembled brick positions $\mathcal{P}$, and brick size $w_b \times d_b$

**Output:** Position of a pivot brick $(a, b, c)$, a relative position of the next brick $(x, y, z)$

1: Initialize a one-initialized brick-sized convolution filter $\mathbf{K} \in \mathbb{R}^{w_b \times d_b \times 1}$, a list of pivot scores $\mathcal{T} = \phi$, and possible relative connections $\mathcal{N}$ with values satisfying the conditions of Equations (8), (9), and (10).
2: Calculate $\mathbf{C}$ using Equation (4).
3: **for** each pivot $(a, b, c) \in \mathcal{P}$ **do**
4:     Calculate $T_{abc}$ using Equation (5).
5:     $\mathcal{T} \leftarrow \mathcal{T} \cup \{T_{abc}\}$.
6: **end for**
7: Sample a pivot brick $(a, b, c) \sim \frac{T_{abc}}{\sum_{T \in \mathcal{T}} T}$.
8: Determine a relative position of the next brick: $(x, y, z) = \arg\max_{(a,b,c) \in \mathcal{N}} \mathbf{C}_{a+x, b+y, c+z}$.

---

**Determining Relative Brick Positions.** After choosing a pivot brick, a relative brick position to the pivot brick is determined to complete the brick placement. Possible relative brick positions $(x, y, z)$ for assembling a $w_b \times d_b$ brick on a pivot brick of size $w_p \times d_p$ are inherently integer-valued positions satisfying following conditions:

$$\left\lfloor \frac{w_b + w_p}{2} \right\rfloor - 1 \leq x \leq \left\lceil \frac{w_b + w_p}{2} \right\rceil - 1, \tag{8}$$

$$\left\lfloor \frac{d_b + d_p}{2} \right\rfloor - 1 \leq y \leq \left\lceil \frac{d_b + d_p}{2} \right\rceil - 1, \tag{9}$$

$$z \in \{-1, 1\}. \tag{10}$$

By considering the conditions described in Equations (8), (9), and (10), we choose the relative brick position $(x, y, z)$ with the highest score of $\mathbf{C}_{t+1}$. A brick must be attached to the pivot brick as we add scores of every attachable brick in Equation (5). In this step, we do not employ a sampling method due to its poor empirical results.

As a result, Algorithm 1 presents the overall procedure to assemble a single brick. By repeating Algorithm 1, we can assemble a number of bricks where either budget-free or budget-aware scenario is assumed.

## 4.2 Training Procedure of the Score Function

Similar to research on language modeling (Mikolov et al., 2010; Sutskever et al., 2014) and reinforcement learning (Sutton & Barto, 2018), our model also predicts a next brick position sequentially. To train such a prediction model, a pair of ground-truth state transition is required as a training sample. However, final voxel occupancy is only available as ground-truth information. It implies that we cannot access intermediate states explicitly. To resolve this issue, we generate an assembly sequence $[\widetilde{\mathbf{B}}_0, \widetilde{\mathbf{B}}_1, \ldots, \widetilde{\mathbf{B}}_{T-1}]$ from the ground-truth voxel occupancy following the procedure described in this section. Eventually, we can utilize this generated sequence to train a model in an autoregressive manner. In addition, generated sequences are unique and diverse, since the stochasticity is injected from the sampling strategy previously introduced.

Since there exist numerous possible sequences to assemble bricks to a certain 3D structure, a single-step look-ahead with a pair of contiguous states, i.e., $(\widetilde{\mathbf{B}}_t, \widetilde{\mathbf{B}}_{t+1})$, is not enough to model practical assembly scenarios. In addition, the training becomes unstable even though the training pairs slightly change. To address these issues, we train our sequential model to predict a $k$-step look-ahead state using pairs of states at step $t$ and step $t+k$, i.e., $(\tilde{\mathbf{B}}_t, \tilde{\mathbf{B}}_{t+k})$. From now, we call this technique *sequence skipping*.

We employ a voxel-wise binary cross-entropy to train our model. By restricting the voxel prediction using a sigmoid function and minimizing the voxel-wise binary cross-entropy, our model learns to predict valid voxel-wise probabilities of the Bernoulli distribution. To sum up, we train our sequential prediction model as follows:

---

**Algorithm 2** Model training for brick assembly

---

**Input:** Dataset $\mathcal{D}$, a batch size $M$, a sequence skipping value $k$
**Output:** Model parameters for brick assembly $\theta$
1:  Generate ground-truth brick assembly sequences $\widetilde{\mathbf{B}}_{0:T}^{(i)}$ and store in $\mathcal{D}_s$ using voxel shapes $x_i \in \mathcal{D}$.
2:  Initialize a buffer $\mathcal{B}$ with $(\widetilde{\mathbf{B}}_{0:T}^{(j)}, x_j, 0)$ where $x_j \sim \mathcal{D}$, $\widetilde{\mathbf{B}}_{0:T}^{(j)} \in \mathcal{D}_s$.
3:  **repeat**
4:      $\mathcal{L} \leftarrow 0$.
5:      Sample and remove a batch $\{(\widetilde{\mathbf{B}}_{0:T}^{(i)}, x_i, t_i)\}_{i=1}^M$ from $\mathcal{B}$.
6:      **for** $(\widetilde{\mathbf{B}}_{0:T}^{(i)}, x_i, t_i)$ in a batch **do**
7:          $\mathcal{L} \leftarrow \mathcal{L} + \log p_\theta(\widetilde{\mathbf{B}}_{t_i+k}^{(i)} | \widetilde{\mathbf{B}}_{t_i}^{(i)})$.
8:          **if** $t_i + k = T$ **then**
9:              Push $(\widetilde{\mathbf{B}}_{0:T}^{(j)}, x_j, 0)$ into $\mathcal{B}$ where $x_j \sim \mathcal{D}$ and $\widetilde{\mathbf{B}}_{0:T}^{(j)} \in \mathcal{D}_s$.
10:         **else**
11:             $\mathcal{B} \leftarrow \mathcal{B} \cup \{(\widetilde{\mathbf{B}}_{0:T}^{(i)}, x_i, t_i + 1)\}$.
12:         **end if**
13:     **end for**
14:     $\theta \leftarrow \theta + \eta \frac{\partial \mathcal{L}}{\partial \theta}$.
15: **until** convergence

---

1. We generate a sequence of voxel occupancy tensors $[\widetilde{\mathbf{B}}_0, \widetilde{\mathbf{B}}_1, \ldots, \widetilde{\mathbf{B}}_T]$ by running our brick assembly method with the ground-truth voxel occupancy in a training dataset;

2. We generate multi-step pairs from $[\widetilde{\mathbf{B}}_0, \widetilde{\mathbf{B}}_1, \ldots, \widetilde{\mathbf{B}}_T]$ in a sliding-window fashion, i.e., $\{(\tilde{\mathbf{B}}_t, \tilde{\mathbf{B}}_{t+k})\}_{t=0}^{T-k}$;

3. We train a transition function $p_\theta(\tilde{\mathbf{B}}_{t+k} | \tilde{\mathbf{B}}_t)$ with $\{(\tilde{\mathbf{B}}_t, \tilde{\mathbf{B}}_{t+k})\}_{t=0}^{T-k}$ and the voxel-wise cross-entropy.

To decorrelate the time steps of training pairs in a batch and shorten training time, we use a buffer throughout training to store training pairs similar to the work (Zhang et al., 2021). The detailed process is described in Algorithm 2.

## 4.3 Inference for Sequential Brick Assembly

In an inference stage, we determine the next brick positions by following our aforementioned procedure, given $\mathbf{B}_0$. We provide a different form of $\mathbf{B}_0$ depending on a task, i.e., completion and generation, and sequentially generate $\mathbf{B}_t$ until a terminal step $T$. For a completion task that is designed to assemble bricks from a incomplete brick structure, we use an intermediate state, i.e., incomplete brick structure, as $\mathbf{B}_0$. For a generation task that is designed to assemble bricks from scratch, we sample an initial brick position from a discrete uniform distribution, i.e., $(x, y, z) \sim \mathcal{U}(\{-2, 2\}^3)$. Then, we assemble bricks on a zero-centered voxel grid of size $(64, 64, 64)$ and use the voxel occupancy of the initial brick position sampled as $\mathbf{B}_0$.

## 4.4 Budget-Aware Sequential Brick Assembly

Suppose that we are given $k$ brick types. Shape of $i$-th brick type is different to the other brick type shapes, where the volume of $i$-th brick type is $v_i$. For the sake of brevity, we define that the volume of the largest brick type is $v_1$ and the volume of the smallest brick type is $v_k$. More concretely, $v_k \leq v_{k-1} \leq \cdots \leq v_1$. A budget of brick types is represented as $\mathbf{c} = [c_1, \ldots, c_k]$ and $\|\mathbf{c}\|_1 = n$ is the total budget of bricks we can assemble, which implies that we have $c_i$ bricks for $i$-th brick type for $i \in [k]$. Then, a probability of the next brick placement $\mathbf{a}_t$ over brick types is post-processed by the following:

$$\bar{p}(\mathbf{a}_t \mid \mathbf{X}_{t-1}, \mathbf{c}_{t-1}) = g\left(\frac{\mathbf{c}_{t-1} \odot \mathbf{v}}{\mathbf{c}_{t-1}^\top \mathbf{v}} \odot p(\mathbf{a}_t \mid \mathbf{X}_{t-1})\right), \tag{11}$$

where $\mathbf{X}_{t-1}$ is a structure that has been assembled until iteration $t-1$, $\mathbf{c}_{t-1}$ is the budget of brick types at iteration $t-1$, $\mathbf{v} = [v_1, \ldots, v_k]$ is the volumes of $k$ brick types, and $g$ is a sum-to-one function. After

choosing the brick type of the next brick and its placement, $\mathbf{X}_{t-1}$ and $\mathbf{c}_{t-1}$ are updated. The rationale behind Equation (11) is that larger and abundant bricks are assembled first. This helps to first build the skeleton of a brick structure with larger bricks and then express the fine parts of the structure. In addition, the consideration of abundance makes abundant bricks consume first.

## 5  Experimental Results

We demonstrate that our model generates diverse structures with high fidelity satisfying assembly constraints in the experiments on the completion and generation of brick structures with distinct brick types. Moreover, more elaborate studies are conducted in order to validate BrECS.

**Dataset.**  To generate ground-truth assembly sequences and the training pairs based on the ground-truth sequences, we use the ModelNet40 dataset (Wu et al., 2015). In particular, the categories of airplane, table, and chair are used for assembly experiments. 3D meshes in the dataset are converted into $(64, 64, 64)$-sized voxel grids, and then they are scaled down to $1/4$ of the original size to reduce the number of required bricks.

**Metrics.**  For the completion task, we use intersection over union (IoU) to evaluate the performance:

$$\text{IoU}(\mathbf{B}^{(1)}, \mathbf{B}^{(2)}) = \frac{\sum_{i=1}^{a} \sum_{j=1}^{a} \sum_{k=1}^{a} [\mathbf{B}^{(1)}]_{ijk} \cap [\mathbf{B}^{(2)}]_{ijk}}{\sum_{i=1}^{a} \sum_{j=1}^{a} \sum_{k=1}^{a} [\mathbf{B}^{(1)}]_{ijk} \cup [\mathbf{B}^{(2)}]_{ijk}}, \tag{12}$$

where $\mathbf{B}^{(1)}, \mathbf{B}^{(2)} \in \mathbb{R}^{a \times a \times a}$. Along with IoU, we measure the ratio of valid structures to all the structures assembled:

$$\% \text{ valid} = \frac{m_{\text{valid}}}{m}, \tag{13}$$

where $m_{\text{valid}}$ is the number of brick structures that satisfy assembly constraints and $m$ is the number of brick structures assembled. In addition, we utilize a class probability of a target class, which is the softmax output of the target class, in experiments on the generation of brick structures. The probability of the target class is measured using a pretrained classifier with the ModelNet40 dataset. We generate 100 samples and report averaged metrics over 100 samples for all experiments.

**Baseline Methods.**  We compare the assembly performance of our method against a sequential assembly method with Bayesian optimization (Kim et al., 2020), denoted as BayesOpt, Brick-by-Brick (Chung et al., 2021), denoted as BBB, and the deep generative model of LEGO graphs (Thompson et al., 2020), denoted as DGMLG, in Tables 2 and 3. BayesOpt optimizes brick positions to maximize IoU between assembled shapes and target shapes. For the method by Kim et al. (2020), we provide exact target structures which belong to a particular category. BBB learns to assemble bricks given multi-view images of target structures. Following its formulation, we also provide three images (top, left, and front) of target structures in a test dataset. DGMLG generates a structure by utilizing the graph representation of brick structures and a deep graph generative model. Note that our approach does not provide any guidance (image or target shape) to produce a new structure. It is noteworthy that such different formulation is inevitable due to their respective assumptions. Importantly, we would claim that our method requires weaker guidance than other methods.

### 5.1  Completion of Brick Structures

We test our method on a completion task for sequential brick assembly where unseen partial structures are given. To establish the completion task for brick assembly problems, we first assemble LEGO bricks using a brute-force approach to filling voxel occupancy in a test dataset with LEGO bricks. Then, we remove a fraction of bricks assembled without losing connectivity between bricks and provide it as an initial state $\mathbf{B}_0$. Each model is trained with a training dataset and then complete brick structures from the initial states. We compare the completion performance by measuring IoU between ground-truth voxel occupancy and complete brick structure. In addition, we report valid assembly ratio and inference time.

As shown in Table 2, our method outperforms the other three baseline methods in terms of IoU. The results show that our method creates high-fidelity brick structures compared to other methods, despite exhaustive

Table 2: Quantitative results of the completion of brick structures. An asterisk after a method name denotes that partial or full ground-truth information is given to the corresponding model.

| Method | IoU (↑) | | | | % valid (↑) | | | | Inference time (sec., ↓) | | | |
|---|---|---|---|---|---|---|---|---|---|---|---|---|
| | airplane | table | chair | average | airplane | table | chair | average | airplane | table | chair | average |
| BayesOpt* | 0.145 | 0.206 | 0.233 | 0.194 | **100.0** | **100.0** | **100.0** | **100.0** | 1.20e6 | 1.11e6 | 1.05e6 | 1.12e6 |
| Brick-by-Brick* | 0.455 | 0.440 | 0.434 | 0.443 | 12.0 | 7.0 | 16.0 | 11.7 | 305.6 | 1502.4 | 2785.2 | 1531.1 |
| DGMLG | 0.315 | 0.269 | 0.271 | 0.285 | 0.0 | 1.0 | 0.0 | 0.3 | 237.3 | 340.0 | 473.0 | 350.4 |
| BrECS (2×4) | 0.571 | 0.586 | 0.534 | 0.564 | **100.0** | **100.0** | **100.0** | **100.0** | **36.3** | **143.9** | **151.0** | **110.4** |
| BrECS (2×4 + 2×2) | **0.599** | **0.594** | **0.541** | **0.578** | **100.0** | **100.0** | **100.0** | **100.0** | 73.8 | 224.1 | 279.0 | 192.3 |

Table 3: Quantitative results of the generation of brick structures with distinct brick types. An asterisk denotes a model with partial or full ground-truth information.

| Method | Class probablity of target class (↑) | | | | % valid (↑) | | | |
|---|---|---|---|---|---|---|---|---|
| | airplane | table | chair | average | airplane | table | chair | average |
| BayesOpt* | 0.039 | 0.043 | 0.069 | 0.050 | **100.0** | **100.0** | **100.0** | **100.0** |
| Brick-by-Brick* | 0.430 | 0.042 | 0.032 | 0.168 | 6.0 | 3.0 | 2.0 | 3.7 |
| DGMLG | 0.228 | 0.023 | 0.027 | 0.093 | 0.0 | 0.0 | 0.0 | 0.0 |
| BrECS (2×4) | 0.415 | **0.250** | 0.404 | 0.356 | **100.0** | **100.0** | **100.0** | **100.0** |
| BrECS (2×4 + 2×2) | **0.447** | 0.229 | **0.419** | **0.365** | **100.0** | **100.0** | **100.0** | **100.0** |

constraint satisfaction. Moreover, our method performs the best in validity ratio alongside BayesOpt, but ours is also the best in inference time. We additionally test our model with distinct brick types by appending 2×2 bricks after assembling 2×4 brick type first. The performance of our method is further improved by using two brick types since different brick types can fill brick positions more densely and express the fine aspect of structures in consequence.

## 5.2 Generation of Brick Structures

As our method is a generative model, our brick assembly model can generate a brick structure that belongs to a particular category. To compare the quality of generated structures semantically, we train a classifier over voxel grids with a small number of 3D convolution layers using the ModelNet40 dataset; the detailed architecture of the classifier is presented in Section F. Given a pretrained classifier over voxel grids, we measure the class probability of generated brick structures for a target class. To feed the voxel grid of generated structure into the classifier, we match the voxel grid size of generated structure with the grid size of the training dataset for the classifier.

Quantitative results are presented in Tables 3 and 4. Our method achieves the best scores in terms of class probabilities and coverage. The results indicate that our method generates diverse and high-quality brick structures compared to the other methods. We also emphasize that the semantic generation quality can be improved using additional 2×2 brick types. Brick structures with distinct brick types are generated by assembling 2×4 bricks first and then 2×2 LEGO bricks. This performance gain is led by the additional improvement on structure refinement that cannot be filled with a single brick type; see Figures 3 and 4.

## 5.3 Budget-Aware Generation

We further conduct experiments on budget-aware scenarios. In the budget-aware experiments, we measure generation performance on various scenarios and analyze how our framework helps to generate 3D structures closer to a target shape. To measure generation quality, we employ the target class probabilities predicted by the pretrained voxel classifier following the evaluation procedure for the generation task.

Table 4: Results on the diversity of generated samples from our method and baselines on 3 different object classes. We generate 100 objects and then measure diversity using the coverage metric similar to the work (Zhang et al., 2021)

| Metric | Class | BayesOpt | Brick-By-Brick | DGMLG | BrECS |
|---|---|---|---|---|---|
| | Airplane | 0.11 | 0.33 | 0.24 | **0.39** |
| Coverage | Table | 0.21 | 0.11 | 0.08 | **0.39** |
| | Chair | 0.34 | 0.26 | 0.22 | **0.44** |

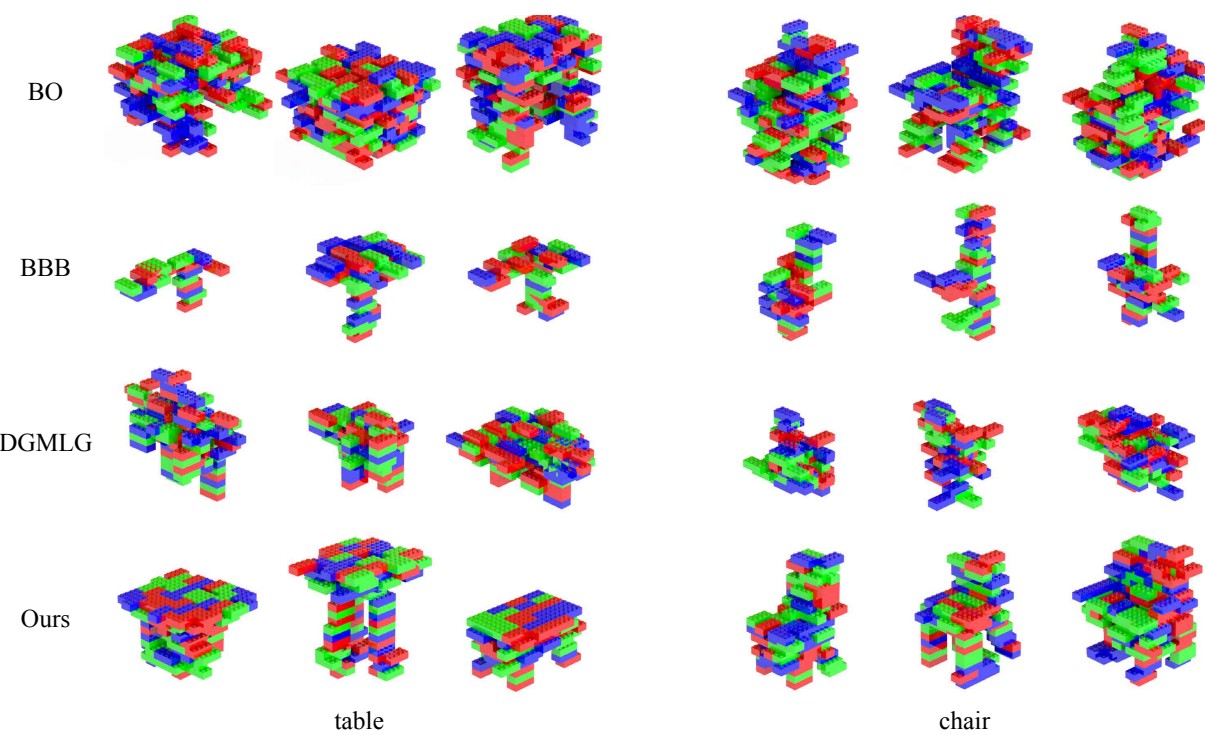

Figure 3: Qualitative results of structure generation. Best viewed in color.

We examine four different scenarios on brick budgets: `even`, `shortage`, `many_big` and `many_small`. To fairly compare them, the same numbers of voxels are provided across the scenarios excluding the `shortage` case; if four 2×4 bricks are given for one scenario, we provide eight 2×2 bricks for another scenario – the numbers of voxels are all 32. We assume that we are given three brick types, i.e., 2×2, 2×4, and 2×8. For the `even` case, we evenly distribute the number of voxels for each brick type; the total number of voxels is 400. For the `shortage` case, we reduce the total number of voxels and evenly distribute the number of voxels for each brick type so that it can fill up to 240 voxels. For the `many_big` case, we allot twice as many voxels to the largest brick type, compared to the other brick types. On the contrary, the number of voxels for the smallest brick type is twice as many as the other brick types in `many_small`.

As shown in Table 5, `even`, `shortage`, and `many_big` outperform 2×4 `only` and `many_small`. It is because the use of diverse brick types can increase the number of possible connections and helps express the fine parts of the structures. Notably, `many_small` fails to create structures since the number of attachable positions for 2×2 bricks is less than the other brick types.

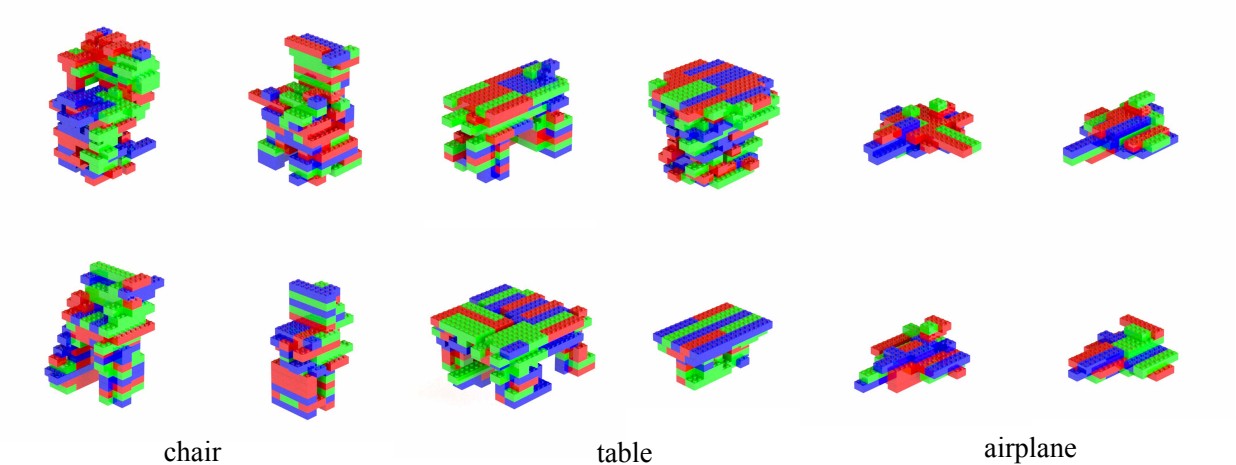

chair                           table                           airplane

Figure 4: Qualitative results of BrECS with $2\times2$, $2\times4$, and $2\times8$ brick types for structure generation. Best viewed in color.

Table 5: Quantitative result of budget-aware brick assembly. A budget is represented as (# $2\times2$ bricks, # $2\times4$ bricks, # $2\times8$ bricks).

| Scenario | # bricks | Max # voxels | Class probability ($\uparrow$) |
|---|---|---|---|
| $2\times4$ only | (0, 150, 0) | 1200 | 0.195 |
| even | (100, 50, 25) | 1200 | 0.329 |
| shortage | (60, 30, 15) | 720 | 0.283 |
| many_big | (50, 25, 50) | 1200 | 0.317 |
| many_small | (200, 25, 12) | 1200 | 0.123 |

## 5.4 Analysis on Our Proposed Model

We analyze the components included in BrECS by verifying each of them in completion or generation tasks, as presented in Tables 6, 7, and 8.

Firstly, we compare different model architectures for a score prediction model. Specifically, we show the comparisons between U-Net and fully convolutional networks. For fair comparisons, we use the same number of convolutional filters. We present the results of generation and completion in Table 6. We find that the U-Net performs better than the fully convolutional network in both generation and completion. We presume that the U-Net is capable of extracting features more robustly because of the pyramidal structure of U-Net.

Secondly, we carry out a study on the impact of stochasticity in the selection process for brick positions by comparing them in terms of completion and generation performance. As presented in Table 7, stochasticity improves generation performance as expected. Interestingly, completion quality is also improved by stochasticity. In the completion task, we remove half of the bricks, and the model completes partially-assembled results, which lose a large amount of their information. Stochasticity may help infer its original shape by making multiple candidates. Wan et al. (2021) also report that stochasticity improves the FID scores of image completion tasks if they are masked out a part of the original image.

Moreover, we compare our original model to models without validity check or sequential skipping in the completion task. As reported in Table 8, each component in our model is effective for improving the quality of brick assembly. Similar to the previous completion experiments, we complete brick structures from the intermediate states of the unseen structures in a test dataset. According to the results, the validity check using convolution filters plays a critical role in constraint satisfaction. Moreover, the performance of our

Table 6: Results of the study on model architectures. The U-Net architecture consistently outperforms the fully convolutional network with the same number of convolutional filters in both completion and generation scenarios.

| Methods | IoU (↑) | | | | Class probability of target class (↑) | | | |
|---|---|---|---|---|---|---|---|---|
| | airplane | table | chair | average | airplane | table | chair | average |
| BrECS with U-Net | 0.571 | 0.586 | 0.534 | 0.564 | 0.415 | 0.250 | 0.404 | 0.356 |
| BrECS with Fully convolutional network | 0.510 | 0.399 | 0.450 | 0.453 | 0.208 | 0.075 | 0.186 | 0.156 |

Table 7: Performance comparisons of BrECS with or without stochasticity. Stochasticity improves performance in both completion and generation tasks.

| Methods | IoU (↑) | | | | Class probability of target class (↑) | | | |
|---|---|---|---|---|---|---|---|---|
| | airplane | table | chair | average | airplane | table | chair | average |
| BrECS with stochasticity | 0.571 | 0.586 | 0.534 | 0.564 | 0.415 | 0.250 | 0.404 | 0.356 |
| BrECS without stochasticity | 0.515 | 0.533 | 0.509 | 0.519 | 0.238 | 0.104 | 0.217 | 0.186 |

Table 8: Results of the study on two components, i.e., validity check and sequence skipping, where we measure completion performance for the chair category. Ours without ablation performs better than ablated variants.

| | IoU | % valid | # bricks | Inference time (sec.) |
|---|---|---|---|---|
| Default setup | 0.534 | 100.0 | 96.7 | 151.0 |
| w/o validity check | 0.441 | 0.0 | 141.6 | 335.8 |
| w/o sequence skipping | 0.437 | 100.0 | 59.3 | 6.0 |

model degrades significantly without sequence skipping as the number of bricks assembled decreases. It implies that the sequence skipping encourages BrECS to predict diverse brick positions.

## 6 Conclusion

We have proposed a brick assembly method to efficiently validate complex assembly constraints and effectively generate high-fidelity brick structures. To sequentially assemble LEGO bricks into 3D structures, our model checks the validity of brick positions using one-initialized brick-sized convolution filters and calculates brick scores utilizing the U-Net architecture. Finally, we showed that our method performs better than several existing methods, tested our method in both budget-free and budget-aware scenarios, and analyzed the components involved in our method through diverse empirical studies.

**Broader Impact Statement**

From the perspective that our model tackles an instance of combinatorial optimization problems, which is an attractive problem in computer science, our work does not have any negative broader impact. However, our approach might be used to generate unethical products, because ours can create novel structures in a combinatorial manner. Thus, this negative ability should be carefully monitored and managed.

**Acknowledgments**

This work was supported by NRF grant (No. 2023R1A1C200781211 (65%)) and IITP grants (RS-2021-II212068: AI Innovation Hub (25%), RS-2021-II211343: AI Graduate School Program at Seoul National University (5%), and RS-2019-II191906: AI Graduate School Program at POSTECH (5%)) funded by the Korea government (MSIT).

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

# A    Overall Procedure of Our Method

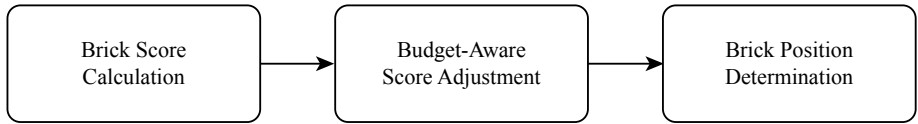

Figure 5: Overall procedure of our method.

Figure 5 presents the overall procedure of our method BrECS. Our method starts from a brick score calculation step with a current brick structure. Then, we adjust brick scores to assemble within a given budget. Finally, we determine a brick position using brick scores from the previous steps. These steps are repeated for assembling a fixed number of bricks into a brick structure.

# B    Details of Implementation

We present the implementation details of baseline methods and our proposed method.

## B.1    Bayesian Optimization

We employ Bayesian optimization to tackle a sequential brick assembly problem. We follow the setup proposed by Kim et al. (2020). The number of bricks is limited to 160 bricks at most. Every brick position is optimized by maximizing IoU between current and ground-truth structures. Note that we have to provide ground-truth voxel information for this strategy.

## B.2    Brick-by-Brick

We use Brick-by-Brick as a baseline method and compare it to our method in terms of the performance of brick assembly. We follow the model architecture and training setup of the model described in the work (Chung et al., 2021). Also, we would like to emphasize that partial ground-truth voxel information have to be provided for this strategy, as the method requires three images of ground-truth shapes to create target shapes. The number of bricks is limited to 75 bricks at most, which is the same as the original setup, due to the excessively increasing memory requirements of the model. In an inference stage, a generation sequence is halted when the newly placed brick violates the constraints, following the environment reset condition of the method.

## B.3    Deep Generative Model of LEGO Graphs

We solve the problem of brick assembly generation using the deep generative model of LEGO Graphs and compare it against ours. We follow the model architecture and training setup described in the work (Thompson et al., 2020). To train the graph generation model, we need the graph representations of target shapes. We therefore create target shapes with ground-truth voxel shapes. Then we convert brick structures into graphs by representing bricks as nodes and direct connections between bricks as edges, following the previous work (Thompson et al., 2020). We train this model for 200 epochs.

## B.4    BrECS

To efficiently train a model for 3D voxel generation, we utilize Minkowski Engine (Choy et al., 2019) and its 3D sparse convolution operation. We train our model with a fixed learning rate of 5e-4, Adam optimizer (Kingma & Ba, 2015), a batch size of 32, sequence skipping with a step size $k = 8$, a buffer size of 1024, and the maximum number of bricks of 150. The input size of our model is $(64, 64, 64)$, and the output size is also $(64, 64, 64)$. We train the model until reaching 100k steps.

## C   Training

We train our model on a server with four NVIDIA GeForce RTX 2080 Ti GPUs.

## D   Details of Hyperparameters

We train our model using the Adam optimizer with a learning rate of 0.0005 and a weight decay of 0.0. We set a batch size to 32, an internal buffer size to 1024, the number of steps to skip to 8, and a voxel size to 64. The numbers of convolution kernels in the score prediction model are 7, 5, 5, 3, 3, 3, 3, 3, 3, and 3 from top to bottom. The number of output channels for the score prediction model is 1.

## E   Detailed Architecture of U-Net

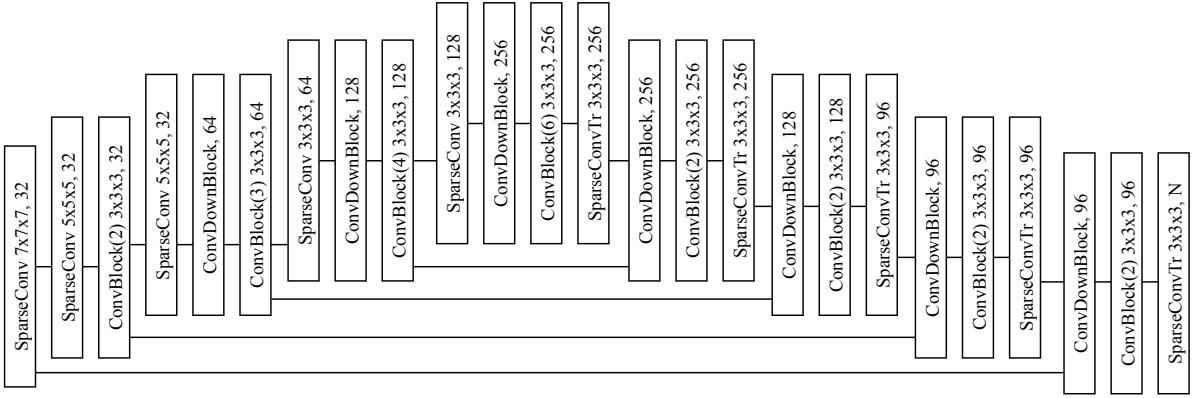

Figure 6: Model architecture of U-Net.

We design our score prediction network inspired by Choy et al. (2019). Detailed model architecture is illustrated in Figure 6.

## F   Detailed Architecture of Voxel Classifier

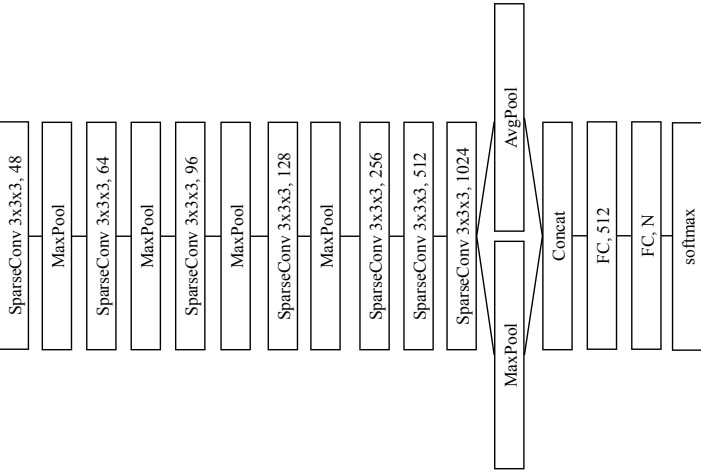

Figure 7: Model architecture of voxel classifier.

We train a shallow voxel classification network to compare several models in terms of generation quality semantically. We add a softmax layer at the end of the network. The detailed network structure is described in Figure 7.

## G Limitations

Our model can process any rectangular bricks with a small amount of modification. However, it is difficult to deal with more diverse brick types such as a brick with a slope and circular brick. Assembling more general brick types and even free-form brick types is left for future work; since non-rectangular materials are common in real-world scenarios, it would make our work more effective. In addition, our model does not consider the inefficiency that occurs when the next brick is attached to multiple bricks previously assembled. More precisely, while different pivot bricks and the corresponding relative positions are selected, the same structure can be produced. In future work, it will be considered to improve assembly efficiency by avoiding such a possibility.

