# OpenReview forum: "Budget-Aware Sequential Brick Assembly with Efficient Constraint Satisfaction"
_TMLR — Accepted by TMLR_

### Review · Reviewer_Tbgf · 2024-06-06

**Summary Of Contributions:**

In this paper, the authors study the problem of sequential brick assembly with LEGO bricks to create 3D structures. They propose to train a predictor to predict 3D voxels after k steps of placing bricks and use it as a score function to determine where to place each brick. They named their method BrECS. They also present a budget-aware version of BrECS that takes remaining brick types and the number of bricks into consideration during construction. Empirical evaluations on the ModelNet40 dataset demonstrate that BrECS achieves higher intersection with the ground truth. BrECS is also better at constructing specific shapes according to a voxel classifier. Finally, the budge-aware version of BrECS is able to adapt to different types of available bricks.

**Audience:**

Yes

**Broader Impact Concerns:**

No concerns.

**Claims And Evidence:**

Yes

**Requested Changes:**

See my questions in the section above. The first 3 weakness points are critical and the next three are relatively minor.

**Strengths And Weaknesses:**

Strengths:
1. The proposed method is a novel one for sequential brick assembly. The main innovation is training a k-step voxel predictor whose predictions are used as scores in selecting the next location to place a brick.

2. There are several other smaller innovations in pivot brick selection and relative brick position determination that are also reasonable. Ablation studies also validated their contribution to BrECS.

3. Empirical evaluations show a decent improvement over baselines in terms of the intersection over union metric.

Weaknesses:
There are several places where the paper is not very clear. The authors should provide more details on BrECS and their experimental setting.

1. In training the k-step look-ahead model, I do not fully understand the data generation method. On page 7 in describing generation for $\tilde{\textbf{B}}_i$, they use ground-truth voxel occupancy to replace $\textbf{B}’$. Does it mean at each step $\textbf{B}’$ stays the same? This is a bit strange.

2. It is unclear to me how this training process provides class information to the model. The empirical evaluations show BrECS is able to construct structures more closely satisfying class labels but I do not see how the model learns to do this.

3. Experimental details are missing the sequence skipping value used for the results. In addition, ablation results on removing sequence skipping show degradation but a more interesting ablation is to measure performances over a range of skipping values.

4. What is the definition of valid bricks in the metric “ratio of valid bricks”?

5. In section 4.2, the authors trained a classifier to measure class probability. How accurate is the classifier?

6. The dataset description could use more details. It seems like the authors used a subset of the ModelNet40 dataset so it would be helpful to provide details on them.

---

> ### Author Response · Authors · 2024-07-07
>
> We sincerely appreciate your valuable feedback.
> You can find our new revision in the submission page.
> In the following, we will answer your concerns and questions.
>
> > In training the $k$-step look-ahead model, I do not fully understand the data generation method. On page 7 in describing generation for $\tilde{\mathbf{B}}_i$, they use ground-truth voxel occupancy to replace $\mathbf{B}'$. Does it mean at each step $\mathbf{B}'$ stays the same?
>
> It is generated in an autoregressive manner. More precisely, a single assembly sequence is split to multiple $k$-length sequences. Therefore, for each sub-sequence, a final output should be matched to the $k$-th element of the sub-sequence. You can imagine the next word prediction or the $k$ next word prediction with sliding windows in language modeling.
>
> We have revised our paper accordingly; please refer to Section 4.2.
>
> > It is unclear to me how this training process provides class information to the model. The empirical evaluations show BrECS is able to construct structures more closely satisfying class labels but I do not see how the model learns to do this.
>
> Similar to previous work on 3D generation, we trained separate models over class labels. We would claim that our method requires weaker guidance than other baseline methods. We have updated the submission accordingly.
>
> > Experimental details are missing the sequence skipping value used for the results.
>
> As presented in Section C of the appendices, we used $k = 8$.
>
> > What is the definition of valid bricks in the metric "ratio of valid bricks"?
>
> We have updated our manuscript. Please see Equation (10).
>
> > In section 4.2, the authors trained a classifier to measure class probability. How accurate is the classifier?
>
> We trained a classifier until validation accuracy is over 90%.
>
> > The dataset description could use more details. It seems like the authors used a subset of the ModelNet40 dataset so it would be helpful to provide details on them.
>
> Thank you for pointing this out. We have revised our manuscript. We highlight the description of the dataset used here.
>
> "To generate ground-truth assembly sequences and the training pairs based on the ground-truth sequences, we use the ModelNet40 dataset. In particular, the categories of airplane, table, and chair are used for assembly experiments. 3D meshes in the dataset are converted into (64, 64, 64)-sized voxel grids, and then they are scaled down to 1/4 of the original size to reduce the number of required bricks."

---

### Review · Reviewer_kiQa · 2024-06-18

**Summary Of Contributions:**

The paper proposes a novel method (BrECS) for sequential LEGO brick assembly in 3 dimensions. BrECS leverages a U-shaped neural network to suggest positions for the next brick, and relies on a convolution operation to efficiently avoid overlaps (the most crucial constraint). It can accommodate different brick types, as well as brick budget constraints. Performance is evaluated on the ModelNet40 dataset and compared to previous algorithms.

**Audience:**

Yes

**Broader Impact Concerns:**

There are no broader impact concerns to address here.

**Claims And Evidence:**

No

**Requested Changes:**

Most of the remarks below are caused by lack of clarity on the fundamental goal of sequential brick assembly. They are ordered by the page I refer to.

- “Supposing that we are given many bricks to assemble” (p. 1) And a target shape?
- “an auxiliary neural network that fails to predict legitimate moves perfectly.” (p. 2) Does your method succeed in perfect prediction?
- “We will release our source code upon publication.” (p. 2) Can you submit it for review?
- “feature” (p. 3) typo
- “Sequential Brick Assembly.” (p. 3) Define the problem more precisely: what is the objective? Matching a given 3d volume?
- “Given a voxel representation of a structure at step t,” (p. 4) Explain what this is
- “Moreover, we expect that our neural network produces a likely complete or potentially next-step 3D structure, which is represented by a probability of voxel occupancy” (p. 5) How do you train it? What is the loss? How does it handle voxels that are already occupied?
- “scores for next brick positions $A_{t+1} \in \mathbb{R}^{a×a×a}$ is computed by sliding a convolution filter $K \in \mathbb{R}^{w_b \times d_b \times 1}$ across $B_t$:” (p. 5) Why is the filter needed here too, and not just for validity?
- “is” (p. 5) typo
- “the size of the convolution filter is the same as the brick size we assemble” (p. 5) Do we need to know ahead of time which brick shape we'll use next?
- “Input: Voxels of structure at current time step $B$, a list of assembled brick positions $P$, and brick size $w_b \times d_b$” (p. 6) What if we don't know which brick will be used next? Does this adapt to heterogenous bricks, possibly with some normalization?
- “$p_{ijk}$” (p. 6) What is this quantity $p_{ijk}$ in Equation (6)?
- “bricks” (p. 6) typo
- “By replacing $B_t$ with ground-truth voxel occupancy, we generate an assembly sequence $[B_0, B_1, \dots , B_{T−1}]$ following the procedure in this section.” (p. 7) This is not clear: how do you recover the intermediate states? How do you ensure that the final ground truth is reached?
- “For a completion task, we use an intermediate state” (p. 7) What is a completion task?
- “For a generation task, we sample an initial brick position” (p. 7) What is a generation task?
- “a probability of the next brick placement is post-processed by the following” (p. 8) What is $\mathbf{a}_t$ in equation (10)?
- “$c_{t−1} \odot v$” (p. 8) In Equation (10), we put more weight on bricks which 1/ are abundant and 2/ have large volume? What is the rationale?
- “Along with IoU, we measure the ratio of valid bricks in structures of interest.” (p. 8) What is the ratio of valid bricks?
- “BayesOpt optimizes brick positions to maximize IoU between assembled shapes and target shapes. BBB learns to assemble bricks given multi-view images of target structures. Following its formulation, we also provide three images (top, left, and front) of target structures in a test dataset.” (p. 8) How can you compare these methods if they use different input data?
- “Note that our approach does not provide any guidance (image or target shape) to produce a new structure.” (p. 8) So it can do whatever it wants?
- “We test our method on a completion task for sequential brick assembly where unseen partial structures are given.” (p. 8) I thought your method didn't require / accept any guidance?
- “Each model is trained with a training dataset and then complete brick structures from the initial states.” (p. 9) What does that mean?
- “Given a pretrained classifier over voxel grids, we measure the class probability of generated brick structures for a target class.” (p. 9) How do you impose the target class on your model?

**Strengths And Weaknesses:**

## Strengths

**Would at least some individuals in TMLR's audience be interested in knowing the findings of this paper? YES.**

The main insight of the BrECS algorithm is that convolutions are very efficient on GPUs, and that they have a natural interpretation for overlap checking. This is a genuinely interesting connection, which yields a fast sequential assembly algorithm.
Numerical results are compelling in some regards, especially the adequation of the generated structure to the target class (which can be assessed visually).

## Weaknesses

**Are the claims made in the submission supported by accurate, convincing and clear evidence? NOT YET.**

My main issue with the paper is that is lacks clarity. The task of "sequential brick assembly" is never properly defined, in particular with respect to a possible target shape. On page 8, the authors state

> Note that our approach does not provide any guidance (image or target shape) to produce a new structure.

But then how do you enforce the generation of something that looks like a chair, or an airplane? How can you compare your method against predecessors in a fair way, when these predecessors work from an exact (Kim et al. 2020) or approximate (Chung et al. 2021) target shape?

It is also unclear to me what the interpretation of the U-shaped neural net should be, and why the convolutional filter $\mathbf{K}$ is a necessary postprocessing step.

As a result of these gray areas, I couldn't assess the experimental results properly. I hope that the authors will be able to explain both points properly in the discussion phase.

---

> ### Author Response · Authors · 2024-07-07
> **Official Comment by Authors (1/n)**
>
> We sincerely appreciate your valuable feedback.
> You can find our new revision in the submission page.
> In the following, we will answer your concerns and questions.
>
> > On page 8, the authors state "Note that our approach does not provide any guidance (image or target shape) to produce a new structure." But then how do you enforce the generation of something that looks like a chair, or an airplane?
>
> > "Note that our approach does not provide any guidance (image or target shape) to produce a new structure." (p. 8) So it can do whatever it wants?
>
> > "Given a pretrained classifier over voxel grids, we measure the class probability of generated brick structures for a target class." (p. 9) How do you impose the target class on your model?
>
> Similar to previous work on 3D generation, we trained separate models over class labels. We would claim that our method requires weaker guidance than other baseline methods. We have updated the submission accordingly. Please refer to Section 5 of the revision.
>
> > It is also unclear to me what the interpretation of the U-shaped neural net should be, and why the convolutional filter $\mathbf{K}$ is a necessary postprocessing step.
>
> As described in Section 4.1, the use of U-Net is for capturing global and local contexts effectively and retaining the same dimensionality. Due to its pyramidal feature extraction structure, the U-Net extracts robust features understanding multi-dimensional contexts.
>
> We have conducted ablation study on this architectural design choice as shown in Table 6. For your convenience, we highlight the ablation study here:
>
> | Methods | IoU - airplane | IoU - table | IoU - chair | IoU - average | CP - airplane | CP - table | CP - chair | CP - average |
> | --- | --- | --- | --- | --- | --- | --- | --- | --- |
> | BrECS with U-Net | 0.571 | 0.586 | 0.534 | 0.564 | 0.415 | 0.250 | 0.404 | 0.356 |
> | BrECS with Fully conv. net. | 0.510 | 0.399 | 0.450 | 0.453 | 0.208 | 0.075 | 0.186 | 0.156 |
>
> As presented in Section 4.1, the application of convolution filters for filtering out invalid brick positions is our primary contribution. With a one-initialized brick-sized convolution filter, invalid positions are easily detected. Please see Figure 2 and Section 4.1 for details.
>
> > "Supposing that we are given many bricks to assemble" (p. 1) And a target shape?
>
> In this context, we have described the difficulty of our problem. The sequential assembly problem has a large number of assemblable brick structures with a sufficient number of bricks to assemble. In addition, in an inference stage, our method assembles bricks without a target shape.
>
> > "an auxiliary neural network that fails to predict legitimate moves perfectly." (p. 2) Does your method succeed in perfect prediction?
>
> Yes, our method achieves 100% of validity with a one-initialized brick-sized convolution filter as shown in Tables 2 and 3.
>
> > "We will release our source code upon publication." (p. 2) Can you submit it for review?
>
> We included our source code in the supplementary material.
>
> > "feature" (p. 3) typo
>
> Thank you for pointing this out. We fixed it.
>
> > "Sequential Brick Assembly." (p. 3) Define the problem more precisely: what is the objective? Matching a given 3d volume?
>
> We have added Section 3 to formulate our problem. Please see Section 3 of the revision.
>
> > "Given a voxel representation of a structure at step $t$," (p. 4) Explain what this is
>
> It is to explain the $t$-th step of brick structure construction. Given a structure at step $t$, our method chooses the next brick position to construct a brick structure at step $t + 1$. As described in the paper, these steps are repeated. You can find the overall procedure of our method in Section A of the appendices.
>
> > "Moreover, we expect that our neural network produces a likely complete or potentially next-step 3D structure, which is represented by a probability of voxel occupancy" (p. 5) How do you train it? What is the loss? How does it handle voxels that are already occupied?
>
> The training details of our method are described in Section 4.2. Briefly speaking, we generated sub-sequences of assembly sequences and used a voxel-wise binary cross-entropy. Voxel occupancy is handled by one-initialized brick-size convolution filters. Please take a look at Section 4.2.

---

> ### Author Response · Authors · 2024-07-07
> **Official Comment by Authors (2/n)**
>
> > "scores for next brick positions $A_{t+1} \in \mathbb{R}^{a \times a \times a}$ is computed by sliding a convolution filter $K \times \in \mathbb{R}^{w_b \times d_b \times 1}$ across $B_t$:" (p. 5) Why is the filter needed here too, and not just for validity?
>
> Using the filter here, it can match to the output of $\mathbf{V}\_t$ in terms of tensor sizes. After calculating $\mathbf{A}\_{t+1}$ and $\mathbf{V}\_t$, they are employed to calculate $\mathbf{C}\_{t+1}$; please see Equation (3).
>
> > "is" (p. 5) typo
>
> We think that there is no typo "is" in Page 5. Please let us know it more specifically.
>
> > "the size of the convolution filter is the same as the brick size we assemble" (p. 5) Do we need to know ahead of time which brick shape we'll use next?
>
> > "Input: Voxels of structure at current time step $B$, a list of assembled brick positions $P$, and brick size $w_b \times d_b$" (p. 6) What if we don't know which brick will be used next? Does this adapt to heterogenous bricks, possibly with some normalization?
>
> No, we do not need to know which brick shape will be used. We applied $k$ convolution filters at the same time where $k$ brick types are considered, and then choose the next brick type and its position choosing the maximum score; please see Equation (8).
>
> > "$p_{ijk}$" (p. 6) What is this quantity $p_{ijk}$ in Equation (6)?
>
> Thank you for pointing this out. It is a redundant expression. We have updated Equation (6).
>
> > "bricks" (p. 6) typo
>
> Thank you for pointing this out. We fixed it.
>
> > "By replacing $B_t$ with ground-truth voxel occupancy, we generate an assembly sequence $[B_0, B_1, \ldots, B_{T-1}]$ following the procedure in this section." (p. 7) This is not clear: how do you recover the intermediate states? How do you ensure that the final ground truth is reached?
>
> It is generated in an autoregressive manner. More precisely, a single assembly sequence is split to multiple $k$-length sequences. Therefore, for each sub-sequence, a final output should be matched to the $k$-th element of the sub-sequence. You can imagine the next word prediction or the $k$ next word prediction with sliding windows in language modeling. It does not need to predict the end of the sequence.
>
> We have revised our paper accordingly; please refer to Section 4.2.
>
> > "For a completion task, we use an intermediate state" (p. 7) What is a completion task?
>
> We have updated our manuscript to specify it.
>
> > "For a generation task, we sample an initial brick position" (p. 7) What is a generation task?
>
> We have updated our manuscript to specify it.
>
> > "a probability of the next brick placement is post-processed by the following" (p. 8) What is at in equation (10)?
>
> We have updated it. Please see the revision.
>
> > "$c_{t - 1} \odot v$" (p. 8) In Equation (10), we put more weight on bricks which 1/ are abundant and 2/ have large volume? What is the rationale?
>
> The rationale behind Equation (8) is that larger and abundant bricks are assembled first. This helps to first build the skeleton of a brick structure with larger bricks and then express the fine parts of the structure. In addition, the consideration of abundance makes abundant bricks consume first. We updated our submission accordingly.
>
> > "Along with IoU, we measure the ratio of valid bricks in structures of interest." (p. 8) What is the ratio of valid bricks?
>
> We have included the description of the ratio of valid bricks. Please see Equation (10) of the revision.
>
> > How can you compare your method against predecessors in a fair way, when these predecessors work from an exact (Kim et al. 2020) or approximate (Chung et al. 2021) target shape?
>
> > "BayesOpt optimizes brick positions to maximize IoU between assembled shapes and target shapes. BBB learns to assemble bricks given multi-view images of target structures. Following its formulation, we also provide three images (top, left, and front) of target structures in a test dataset." (p. 8) How can you compare these methods if they use different input data?
>
> Target structures in a test dataset are used for BayesOpt and BBB and their performances are measured. We included the following sentences for clarity:
>
> "It is noteworthy that such different formulation is inevitable due to their respective assumptions. Importantly, we would claim that our method requires weaker guidance than other methods."
>
> > "We test our method on a completion task for sequential brick assembly where unseen partial structures are given." (p. 8) I thought your method didn't require / accept any guidance?
>
> This is not guidance. More specifically, we do not provide a final target structure. Unlike our method, BayesOpt and BBB require exact target structures and partial information of exact target structures, respectively. It is analogous to a completion task in computer vision. The goal of this task is to complete a given structure.

---

> ### Author Response · Authors · 2024-07-07
> **Official Comment by Authors (3/n)**
>
> > "Each model is trained with a training dataset and then complete brick structures from the initial states." (p. 9) What does that mean?
>
> It means that we trained our method with a training dataset and assembled bricks to $\mathbf{B}_0$. $\mathbf{B}_0$ is $\mathbf{B}$ at step 0, i.e., an initial state; please see Algorithm 1.

---

> ### Comment · Reviewer_kiQa · 2024-07-29
> **Answer to authors**
>
> Thank you for your clarifications. I will not go back to them one by one but instead try to summarize my remarks by theme. Some are prompted by your answers, some by my second reading of the paper.
>
> **About the assembly problem itself (section 3)**
>
> When I read "sequential brick assembly", the first thing I have in mind is the typical LEGO kit which must be used to reproduce a structure _exactly_. If I buy a LEGO airplane and follow the instructions, the end result will be precisely the one I see on the box, with 100% overlap.
>
> But from what I understand, you never try to achieve this. Instead you train a model to "build something that looks like an airplane", and then let it improvise either from a partial structure or no structure at all. This means that you cannot steer it towards a precise end result, unlike predecessor methods which leverage the ground truth. Is it correct that the full target volume is never part of the input at inference time, and is only used during training?
> If so, you're right that this method requires less supervision than predecessors. However, it also means the comparison is a bit unfair, especially on the fully generative task (Table 3).
>
> In any case, it would be helpful to describe the goal / objective of the problem you consider in section 3, not just the constraints.
>
> **About the neural network (section 4.1)**
>
> The part that remains unclear to me is the reason why $K$ appears twice in your pipeline of Figure 1. I understand its role for validity checking, but I don't see why it also has to be appended to the U-shaped neural network. My hypothesis is that the U-Net be interpreted as giving scores to bricks -inside the existing structure $B_t$, and it's up to the convolution $K$ to spread these scores to nearby voxels. Is that right? Why can't the neural network directly predict scores for adjacent voxels without convolution?
>
> **About brick placement (section 4.1)**
>
> Once the scores are computed, you choose a single pivot brick $(a, b, c)$ and a single displacement $(x, y, z)$. But oftentimes the new brick will be attached to 2 or 3 existing bricks. In addition, there are several couples of pivot brick and displacement that can lead to the same physical result, e.g. $(a-1, b, c)$ and $(x+1, y, z)$. Can you explain how your model deals with such degeneracy?
>
> The "is" typo I mentioned is in the first sentence of the second paragraph of page 5: "scores for next brick positions is [sic] computed".

---

> > ### Comment · Reviewer_kiQa · 2024-07-29
> > **Answer to authors (3)**
> >
> > Please keep in mind that I do find the paper interesting and novel. My only goal is to help improve it for the many readers of TMLR who, like me, have never heard of sequential brick assembly before.
> >
> > For the editor: at the moment, my main remaining concerns are on
> >
> > - The clarity of the exposition
> > - The fairness of the evaluation
> >
> > I eagerly await clarifications from the authors, and apologize for the delay in my review.

---

> > > ### Author Response · Authors · 2024-07-30
> > > **Follow-up Comment by Authors (1/n)**
> > >
> > > > When I read "sequential brick assembly", the first thing I have in mind is the typical LEGO kit which must be used to reproduce a structure exactly. If I buy a LEGO airplane and follow the instructions, the end result will be precisely the one I see on the box, with 100% overlap. But from what I understand, you never try to achieve this. Instead you train a model to "build something that looks like an airplane", and then let it improvise either from a partial structure or no structure at all. This means that you cannot steer it towards a precise end result, unlike predecessor methods which leverage the ground truth. Is it correct that the full target volume is never part of the input at inference time, and is only used during training?
> > >
> > > Yes, you are correct. The full target volume is never part of the input at inference time and is only used during training, unlike the previous methods.
> > >
> > > > If so, you're right that this method requires less supervision than predecessors. However, it also means the comparison is a bit unfair, especially on the fully generative task (Table 3).
> > >
> > > Please note that our method solves a more difficult problem because it uses less supervision than predecessors. Even though less supervision is provided, our method shows better performance in terms of the metrics used in this work. Moreover, we have clearly described the details of the comparisons conducted in our manuscript.
> > >
> > > > In any case, it would be helpful to describe the goal / objective of the problem you consider in section 3, not just the constraints.
> > >
> > > We added the description of the goal of the problem we consider. Please see Section 3.
> > >
> > > > The part that remains unclear to me is the reason why $K$ appears twice in your pipeline of Figure 1. I understand its role for validity checking, but I don't see why it also has to be appended to the U-shaped neural network. My hypothesis is that the U-Net be interpreted as giving scores to bricks -inside the existing structure $B_t$, and it's up to the convolution $K$ to spread these scores to nearby voxels. Is that right? Why can't the neural network directly predict scores for adjacent voxels without convolution?
> > >
> > > Your understanding is correct. The output of the U-shaped neural network is voxel-wise scores, so nearby voxels can be aggregated by a brick-sized convolution filter. After applying the convolution filter, each position of the output corresponds to a placement score when that position is selected. When we consider different brick types, the application of brick-sized convolution filters to obtain placement scores is more reasonable than predicting scores without convolution filters, as we need different scores for different brick types.
> > >
> > > > Once the scores are computed, you choose a single pivot brick $(a, b, c)$ and a single displacement $(x, y, z)$. But oftentimes the new brick will be attached to 2 or 3 existing bricks. In addition, there are several couples of pivot brick and displacement that can lead to the same physical result, e.g. $(a-1, b, c)$ and $(x+1, y, z)$. Can you explain how your model deals with such degeneracy?
> > >
> > > You pointed out an important point. Indeed, we did not consider the possibility of our method's inefficiency. We described it in Section G (Limitations) of the revision.
> > >
> > > > The "is" typo I mentioned is in the first sentence of the second paragraph of page 5: "scores for next brick positions is [sic] computed".
> > >
> > > Thank you! We fixed it.
> > >
> > > > If my understanding of your method is accurate, the ground truth cannot be used to steer construction towards a precise structure. Therefore, I cannot fathom how you proceed to generate your training data by decomposing a ground truth $B\_T$ into assembly steps $\widetilde{B}\_t$ so that the last step $\widetilde{B}\_T$ equals the ground truth? How do you enforce it without any backtracking? If you don't enforce this equality, how is the ground truth used at all?
> > >
> > > As described in the previous response, please imagine an autoregressive model for language modeling and computer vision. Our model predicts the structure of the next $k$ steps, so it does not need backtracking or any enforcement.
> > >
> > > > As far as the loss function is concerned, can you specify between which objects the voxel-wise cross-entropy is taken?
> > >
> > > As described in Section 5, objects are taken from the ModelNet40 dataset. The voxel representations of objects in the ModelNet40 are used to calculate the voxel-wise cross-entropy.

---

> > > > ### Author Response · Authors · 2024-07-30
> > > > **Follow-up Comment by Authors (2/n)**
> > > >
> > > > > And how the transition function written $p\_\theta$ relates to the U- Net's parameters?
> > > >
> > > > The output of our network corresponds to the probabilities of all positions since a sigmoid function is applied. Then, our network, which is parameterized by $\theta$, is trained by the binary cross-entropy.
> > > >
> > > > > The dataset generation you describe leaves some areas blurry. "we first assemble LEGO bricks using voxel occupancy in a test dataset." How do you do this? Using your own model?
> > > >
> > > > We revised it as "we first assemble LEGO bricks using a brute-force approach to filling voxel occupancy in a test dataset with LEGO bricks."
> > > >
> > > > > "we remove a fraction of bricks assembled without losing connectivity between bricks and provide it as an initial state" How are these bricks chosen? How big is the fraction?
> > > >
> > > > We randomly removed 50% of the bricks assembled from the structures assembled without losing connectivity.
> > > >
> > > > > I don't understand what the "number of valid bricks" in a structure means. After all, if they're not valid, they cannot be assembled. Do you mean the number of valid bricks among all those proposed by a model, knowing that you discard infeasible propositions? Otherwise I cannot interpret the meaning of the 141.6 in table 8, knowing that 0% of the bricks are valid.
> > > >
> > > > While they are not valid and they cannot be assembled, we can predict brick placements. For Table 8, since some bricks assembled at the beginning start to violate the constraints, our model fails to predict valid brick positions. We presume that our model has collapsed due to unseen invalid placements.
> > > >
> > > > > You also say that, for Brick-by-Brick, generation is halted as soon as an invalid brick is suggested. Is that how the original authors evaluated it? Why wouldn't you just try again?
> > > >
> > > > Yes, we contacted the original authors and conducted the experiments following them.

---

> ### Comment · Reviewer_kiQa · 2024-07-29
> **Answer to authors (2)**
>
> **About training (section 4.2)**
>
> If my understanding of your method is accurate, the ground truth cannot be used to steer construction towards a precise structure. Therefore, I cannot fathom how you proceed to generate your training data by decomposing a ground truth $B_T$ into assembly steps $\widetilde{B}_t$ so that the last step $\widetilde{B}_T$ equals the ground truth? How do you enforce it without any backtracking? If you don't enforce this equality, how is the ground truth used at all?
>
> As far as the loss function is concerned, can you specify between which objects the voxel-wise cross-entropy is taken? And how the transition function written $p_\theta$ relates to the U-Net's parameters?
>
> **About completion (section 5.1)**
>
> The dataset generation you describe leaves some areas blurry.
>
> > we first assemble LEGO bricks using voxel occupancy in a test dataset.
>
> How do you do this? Using your own model?
>
> > we remove a fraction of bricks assembled without losing connectivity between bricks and provide it as an initial state
>
> How are these bricks chosen? How big is the fraction?
>
> **About performance evaluation (section 5 and appendix)**
>
> I don't understand what the "number of valid bricks" in a structure means. After all, if they're not valid, they cannot be assembled. Do you mean the number of valid bricks among all those proposed by a model, knowing that you discard infeasible propositions? Otherwise I cannot interpret the meaning of the 141.6 in table 8, knowing that 0% of the bricks are valid.
>
> You also say that, for Brick-by-Brick, generation is halted as soon as an invalid brick is suggested. Is that how the original authors evaluated it? Why wouldn't you just try again?

---

### Review · Reviewer_VkFp · 2024-07-04

**Summary Of Contributions:**

The paper aims to solve the problem of sequential assembly of brick to create combinatorial 3D structures. In particular, the paper proposes to new method that first, predict the scores of the next brick position using a U-shaped sparse 3D convolutional neural network, second, use a convolution filter to validate physical constraints between bricks, and finally, use a sampling strategy to determine the next brick position based on the physical constraints. The proposed method is evaluated on the ModelNet40 dataset, and was compared various baselines.

**Audience:**

Yes

**Claims And Evidence:**

Yes

**Requested Changes:**

[1] More motivation for the components in the proposed method (e.g., the use of U-shaped sparse 3D convolutional neural network or the convolution filter)

[2] Explanations of why only one dataset is used to evaluate all the methods (if there are more datasets available, need to evaluate with more datasets). Descriptions of all the metrics used in the experimental section.

**Strengths And Weaknesses:**

Strengths:
+ The paper is very well-written, all the concepts are explained clearly, and the proposed method implementation is also described in detail.
+ The general idea of the proposed method seems to be reasonable (although I have various concerns below regarding the detailed implementation of the proposed method).
+ The experiments show the proposed method outperforms existing techniques by a very high margin, and across different aspects.

Weaknesses:
+ I think the motivation of the components of the proposed method could be elaborated much more. For example, when predicting the next brick position, the use of U-shaped sparse 3D convolutional neural network needs to be explained in more detail. The paper only said it is inspired by Choy et al. (2019) but no clear rationale is given. Why don’t we use another type of convolutional neural network? What makes U-shaped one work well in this case? For the filtering invalid brick positions, why do we just apply a convolution filter? Why don’t we use a U-shaped sparse 3D CNN as in the 1st component?
+ For the experimental evaluation, I’m wondering why only one dataset (ModelNet40) is used. Is there only one dataset for this type of problem? And for the metrics, in the beginning of Section 4, the paper only describes the IoU metric, but then later, the paper also evaluates all the methods with other metrics like coverage, but the description of the coverage metric is not described in detail. And furthermore, because we have multiple metrics, what metrics are more important in this type of problem?

---

> ### Author Response · Authors · 2024-07-07
>
> We sincerely appreciate your valuable feedback.
> You can find our new revision in the submission page.
> In the following, we will answer your concerns and questions.
>
> > More motivation for the components in the proposed method (e.g., the use of U-shaped sparse 3D convolutional neural network or the convolution filter)
>
> > I think the motivation of the components of the proposed method could be elaborated much more. For example, when predicting the next brick position, the use of U-shaped sparse 3D convolutional neural network needs to be explained in more detail. The paper only said it is inspired by Choy et al. (2019) but no clear rationale is given.
>
> As described in Section 4.1, the use of U-Net is for capturing global and local contexts effectively and retaining the same dimensionality. Due to its pyramidal feature extraction structure, the U-Net extracts robust features understanding multi-dimensional contexts.
>
> > Why don’t we use another type of convolutional neural network? What makes U-shaped one work well in this case?
>
> We have conducted ablation study on this architectural design choice as shown in Table 6. For your convenience, we highlight the ablation study here:
>
> | Methods | IoU - airplane | IoU - table | IoU - chair | IoU - average | CP - airplane | CP - table | CP - chair | CP - average |
> | --- | --- | --- | --- | --- | --- | --- | --- | --- |
> | BrECS with U-Net | 0.571 | 0.586 | 0.534 | 0.564 | 0.415 | 0.250 | 0.404 | 0.356 |
> | BrECS with Fully conv. net. | 0.510 | 0.399 | 0.450 | 0.453 | 0.208 | 0.075 | 0.186 | 0.156 |
>
> > For the filtering invalid brick positions, why do we just apply a convolution filter? Why don’t we use a U-shaped sparse 3D CNN as in the 1st component?
>
> As presented in Section 4.1, the application of convolution filters for filtering out invalid brick positions is our primary contribution. With a one-initialized brick-sized convolution filter, invalid positions are easily detected. Please see Figure 2 and Section 4.1 for details.
>
> > Explanations of why only one dataset is used to evaluate all the methods.
>
> The ModelNet40 dataset is widely used in the 3D vision field, because it contains many realistic objects. We improve the description of the dataset we used as follows:
>
> "To generate ground-truth assembly sequences and the training pairs based on the ground-truth sequences, we use the ModelNet40 dataset. In particular, the categories of airplane, table, and chair are used for assembly experiments. 3D meshes in the dataset are converted into (64, 64, 64)-sized voxel grids, and then they are scaled down to 1/4 of the original size to reduce the number of required bricks."
>
> > Descriptions of all the metrics used in the experimental section.
>
> We have included the description of all the metrics. Please see Equations (9) and (10) and Section 5 of the revision.

---

### Author Response · Authors · 2024-07-07

Dear Reviewers and Action Editor,

We thank you for your productive comments. Considering your reviews, we have revised our submission. Moreover, we have answered your concerns and questions in individual replies. If you have any follow-up questions, please let us know.

---

### Decision · Action_Editor_yf38 · 2024-09-04

**Recommendation:** Accept with minor revision

**Comment:**

Two of the three reviewers indicated that this initial submission already met the TMLR criteria. The reviewers had some suggestions and questions, which the authors addressed in their response and updates to the paper.

The third reviewer also recommended acceptance after a comprehensive discussion with the authors. What came out of the discussion is that there are aspects of the paper that the authors might wish to expand upon to improve the paper's reproducibility and make it more accessible, especially to researchers somewhat outside this subfield. In particular, based on the reviewer's suggestions here are the minor modifications I suggest:

-   Exposition. The reviewer asks "to clarify how the training dataset is generated: how a ground truth volume is decomposed into individual brick additions" My understanding from the paper is that this is already provided in Section 4.2. If this is the case, no additional modification is required.
- Exposition. Your new Section 3 is helpful, but I support the reviewer's suggestion to also specify/discuss the objective of sequential brick assembly for readers.
-  Reproducibility. You write "We will release our source code upon publication." Providing the method code and the code to generate the datasets would be most helpful.

I am happy to recommend acceptance, congratulations!

**Audience:**

All reviewers also agree that some individuals in the TMLR audience would be interested in this work.

The reviewers also suggest a few relatively minor changes that would make this paper more accessible to people outside of this subfield. I detail those in my comments below.

**Claims And Evidence:**

Following the reviewer discussion, all reviewers agree that the claims are correctly supported by evidence.

---

> ### Author Response · Authors · 2024-09-24
>
> Dear Action Editor,
>
> We have uploaded the camera-ready version of our work and the GitHub link to our implementation by reflecting the Action Editor's and Reviewers' comment.
>
> Please let us know if our manuscript needs any further updates.